# A study on the purchase intention of luxury goods from the perspective of face perception and expected regret

**Jiahua Wei** [ORCID] *, **Hong Shen, Zhizeng Lu**

Business School, Guilin University of Technology, Guilin, Guangxi, China

* jiahua6688@163.com

**Data Availability Statement:** All relevant data are within the manuscript and its Supporting Information files.

**Funding:** Funding 1. Guangxi Natural Science Foundation (Grant No. 2021GXNSFAA220066). 2.

## Abstract

The study on the impact of consumer purchase intention on luxury goods has received widespread attention from the academic community. This study collected research data in Guilin, China, through questionnaire survey, and conducted an empirical study on the influencing factors of luxury consumers' purchase intention. The results show: The price level of luxury goods has a positive impact on consumers' face perception, while the positive impact of price level on expected regret has not been verified. Consumer's face perception has positive and negative effects on consumers' expected regret and consumers' purchase intention respectively. Consumer's downward expected regret and consumer's upward expected regret have different effects on consumers' purchase intention. Consumers' face perception and expected regret play a mediating effect in the research of influence relationship. This study is conducive to a better analysis of the psychology and behavior of Chinese luxury consumers, enriching the theoretical connotation of consumer psychology, and promoting the healthy development of the luxury goods industry.

## Introduction

In 2023, the global economy continued to decline, and the luxury goods market overall showed a downward trend. On October 24, 2023, Kering in France released its performance report for the third quarter of 2023, with sales falling 13% year-on-year. However, the Chinese luxury goods market has become an important engine for the entire industry. Morgan Stanley pointed out in his article "Chinese Consumers Boost the Luxury Industry" that in 2023, Chinese consumers will further drive demand for high-end clothing, accessories, and other luxury goods. The Chinese market has become an important source of income for most luxury brands. In the current era of weak demand in the global luxury goods market, seizing Chinese consumers means seizing opportunities. Therefore, it is very important to study the purchasing intention of luxury consumers. How does consumer willingness to purchase arise? What factors will affect consumers' willingness to purchase? This is a practical issue proposed in the current luxury consumption field.

For a long time, due to the strong demand for luxury goods in China, many scholars have paid attention to the psychology and behavior of luxury goods consumption. Some scholars

National Natural Science Foundation of China (Grant No. 72262009). The funders will provide funding for research design, data collection, and analysis in this paper, and will support the publication of the paper.

**Competing interests:** The authors have declared that no competing interests exist.

believe that Chinese people's face perception is the main internal driving force of luxury consumption, and its impact on the purchase intention of "difficult to obtain luxury goods", "available luxury goods" and "new luxury goods" [1,2]. In the consumption of luxury goods, price comparison would have a significant positive impact on face [3]. At present, the relationship between price and expected regret, as well as the relationship between expected regret and purchase intention has also been involved in academic circles [4,5], but these studies are still messy, and have not formed a systematic theoretical system, and rarely involve the field of luxury consumption. In luxury goods consumption, consumers' face perception and expected regret are objective, and the price of luxury goods may also affect consumers' expected regret, but academic research is still lacking.

Therefore, this study used a questionnaire survey method to collect research data in Guilin City, China, and conducted empirical research. This study attempts to explore the relationship between price level, face perception, expected regret, and consumer purchase intention in the context of luxury consumption in China, and to test the mediating effect of face perception, expected regret, and consumer purchase intention between price level and consumer purchase intention. This study is based on the luxury consumption scenario in China and delves into the above issues. It will expand the applicability of psychological variables such as face perception and expected regret in luxury consumption research, better analyze the psychology and behavior of Chinese luxury consumers, and enrich the connotation of consumer psychology.

## 2. Literature review and research hypothesis

### 2.1 Face perception and price level

In Chinese culture, face reflects the relationship basis between people to a considerable extent. Face, fate and human feelings are regarded as the three goddesses ruling China, and human feelings and face are the two core elements to understand Chinese behavior [6]. Face is related to personal image and reputation. Therefore, Chinese people's understanding of face determines whether they challenge tradition to earn face or reserve opinions to avoid losing face when making innovation decisions [7]. Face is not only related to personal social status, but also related to others' recognition of themselves. From the sociological point of view, face is the overall evaluation of social value, prestige or respect given to individuals by society; From a psychological perspective, face is an individual's internal self-esteem and self-evaluation of social image [8]. Based on the concept of face from the subjective and objective perspectives, face can be viewed as an objective external evaluation of one's social image from others, as well as an individual's subjective perception of self-esteem and social recognition image, and as an individual's self- projection in the social field [9]. Face orientation is an extension of the connotation of face, which is a psychological expression of Chinese people's behavior based on their sensitivity to face [8]. As for the dimension research of face, there are two dimensions, three dimensions and four dimensions. The two-dimensional view divides face into social face and moral face [6]. Face can be divided into three dimensions: Ability of based face, relationship of based face and moral of based face needs [9]. Face can be divided into four dimensions: Moral face, ability face, status face, and relationship face [1]. In the conflict between superiors and subordinates, members with high power status in the culture of small power distance may tend to use direct face conflict management strategies, such as direct disapproval strategies and autonomous threat strategies to induce members with low power status [10]. Human feelings are closely related to face. Generally speaking, there must be face when there is human kindness. To give face is to give it away, so Chinese people often call them "face" together [11]. Influenced by China's collectivist values, Chinese consumers will pay special attention to face when clarifying their purchase intentions, and like "conspicuous consumption" to win the

admiration and appreciation of others [12]. Overall, face as an important topic in interpersonal communication and human practice, has attracted the attention of researchers in different disciplines for a long time. Based on different academic perspectives, they discussed the connotation and extension of face, and have formed a series of theories so far.

In the hotel industry, people living in lower priced rooms are more concerned about exploring the convenience of the surrounding area, while those living in higher priced rooms are more concerned about the surrounding environment and internal facilities of the property [13]. From the perspective of marketing that face will affect consumers' attitudes and purchase behavior, and believed that research showed that face threat will have a positive impact on consumers' impulsive purchase [14]. There is study showed that the concept of face played a mediating effect in the process of self-concept influencing wine purchase intention [15]. There are also studies that show that price comparison will have a significant positive impact on face, and it will have a stronger positive impact on face in product categories with stronger ability to show identity (such as luxury goods), as well as in consumers with higher materialism [3]. Therefore, when the price level of luxury goods increases, consumers' self-esteem and social image will also be improved, that is, face perception will be improved. Therefore, the following research assumptions are proposed:

H1: The price level of luxury goods has a significant positive impact on consumers' face perception.

## 2.2 Expected regret

Because there is not only a choice in life, regret is a regular psychological behavior, which is a human instinct. If individuals realize that they can make other choices in advance and the results will be better, regret will occur. Regret includes experience regret and expected regret. Expected regret refers to anxiety, hesitation and suspicion caused by fear of possible losses before making a choice [5]. Crawford study suggests that people do not spontaneously anticipate the regrets they may experience in the affected situation. In addition, when asked to predict such regrets, they may have misunderstandings about their future feelings [16]. The smaller the relative intensity of expected regret, the higher the optimal price, and the higher the optimal profit [17]. Consumers' expected regret in the service industry would increase the threshold for online merchants to participate in market competition, and the value of service strategies was positively related to the sensitivity of expected regret [18]. The research also shows that expected regret has a significant effect on consumer purchase behavior, and under different expected regret intensity, enterprises will achieve equilibrium conditions [17]. At the same time, the direction of expected regret will also directly affect impulsive buying behavior [5]. The expected regret factor of consumers, established a product promotion strategy model for enterprises in the market composed of two enterprises that provide new and old products respectively, and analyzed the impact of expected regret on product promotion, pricing strategies and profits of enterprises [4].

In the price strategy, when the availability of products in the liquidation period is low, if the expected regret is not considered, the enterprise always chooses the commitment pricing strategy. If expected regret is considered and consumers are more sensitive to high price regret, enterprises will adopt dynamic pricing strategies. When the availability of products in the liquidation period is high, there is no difference between the two pricing strategies [19]. Zhang et al. confirmed that no matter what kind of pricing strategy an enterprise adopts, all consumers who buy existing products will buy new products by exchanging the old for the new [4]. While enterprises prefer to adopt penetration pricing strategy. When the relative intensity of

consumers' expected regret is low and the residual value of old products is relatively large, enterprises can obtain higher profits by selling old products without reducing prices. In the luxury goods industry, the price elasticity of various commodities is relatively high, and price changes will cause greater changes in sales. Therefore, if the price of luxury goods increases, consumers will make more urgent decisions about whether to buy. However, when the price of luxury goods increases, the expected regret will increase, while when the face perception increases, the expected regret will decrease. Therefore, the following research assumptions are proposed:

H2: The price level of luxury goods has a significant positive impact on consumers' expected regret.

H3: Consumers' face perception has a significant negative impact on their expected regret.

## 2.3 Consumers' purchase intention

In the retail industry, the opportunities for retailers to obtain green finance have a positive impact on consumers' willingness to purchase green products [20]. In addition, in sales activities, virtual CSR co creation will promote customers' willingness to purchase green products [21]. There is currently research exploring the interaction between the certification, bidding, and service effort decisions of freelancers and the design and award of outsourcing contracts by clients [22]. The impact of information posts and storytelling posts, and found that when brand image is not enhanced, consumers perceive influential individuals as more authentic, on the contrary, consumers have a more positive attitude towards posts and recommended products after reading them [23].

Some scholars have studied the relationship between consumers' face perception and purchase intention. When consumers gain face, they will have positive emotions, which will lead to impulsive buying behavior. Consumers with high face values will have more positive emotions, while consumers with low self-esteem will have more obvious impulsive buying behavior after gaining face [24]. Although there were differences between groups with moral face awareness and groups with relational face awareness in their responses to different advertising information, both had a significant impact on their purchase intentions [2]. Chinese people have obvious characteristics of others' orientation. In the process of interaction with society, face plays a very important role. Informative social influence tendency will indirectly affect impulse buying through the intermediary variable of face protection tendency [25]. The characteristics of consumers' sense of face obligation and others orientation have a positive and significant impact on the purchase intention of "difficult to obtain luxury goods" [1]. Jeon confirmed through experimental research that people believe that the impact of expected regret on unplanned purchasing behavior is influenced by self-control tendencies. However, self-control has little effect on regulating variables [26]. When purchasing luxury goods, people's face perception will be enhanced, which will enhance their purchase intention. As for the research on expected regret and purchase impact and behavior, some researchers believe that consumers' expected regret direction has a significant impact on impulsive purchase behavior, that is, consumers' downward expected regret (if they don't buy, they will regret) will be more likely to have impulsive purchase than upward expected regret (if they buy, they will regret) [5]. If consumers feel that they will regret not buying, it is expected that regret will increase their willingness to buy luxury goods, and vice versa. Therefore, the following two research assumptions are proposed:

H4: Consumers' face perception has a significant positive impact on consumers' purchase intention.

H5a: Consumers' downward expected regret will have a significant positive impact on consumers' purchase intention;

H5b: Consumers' upward expected regret will have a significant negative impact on consumers' purchase intention.

At the same time, from the above theoretical analysis and research assumptions, we can see that face perception and expected regret play a mediating effect between price level and consumers' purchase intention respectively. In addition, this study also has a chain mediating effect. In the relationship between price level and consumers' purchase intention, face perception and expected regret play a chain mediating effect. Therefore, the following three research hypotheses of mediating effect are proposed:

H6: Face perception plays a mediating effect between price level and consumers' purchase intention;

H7: Expected regret plays a mediating effect between price level and consumers' purchase intention;

H8: In the relationship between price level and consumers' purchase intention, face perception and expected regret play a chain mediating effect.

Based on the above research assumptions, this paper constructs a research model figure as shown in Fig 1.

## 3. Research method

### 3.1 Questionnaire design

In this paper, a pre-survey questionnaire is designed and carried out. The pre-survey includes four variables involved in the research model: price level, face perception, expected regret and purchase intention. Each variable consists of 3–5 items, 16 items in total. The questionnaire was measured with Likert 5-point scale, and the options ranged from "very consistent, consistent, basically consistent, not very consistent" to "not consistent", which was divided into 5 grades.

In pre-survey, a total of 86 questionnaires were distributed and 73 valid questionnaires. Because Cronbach's α less than 0.70, two items are deleted, namely "I hope everyone knows I have money" under the face perception variable and "I am worried about exceeding the family expenditure budget" under the expected regret variable. There are 14 main items in the final formal questionnaire, with 4 items under each variable. In addition to the main items, the questionnaire also includes the gender, age, education background and monthly income of the respondents.

### 3.2 Formal questionnaire

The formal questionnaire was conducted in Guilin, a famous tourist city in China. The survey period is from January 2, 2023 to January 10, 2023. At that time, residents from various parts of China were relatively free to go shopping and move across regions.

The survey was carried out at the Chow Tai Fook Jewelry Store, Luxury Goods Living Hall, Jin Tai Fook Jewelry Store, and the jewelry counter of the Xiaoxiao Hall Commercial Building in Guilin. This survey is aimed at consumers. The questionnaire is distributed on the spot and collected on the spot. During the filling process, the researcher explained the dimensions and items of the questionnaire due to the concern that the respondents had insufficient understanding of the questionnaire. In addition, the researcher allows the respondents to fill in the

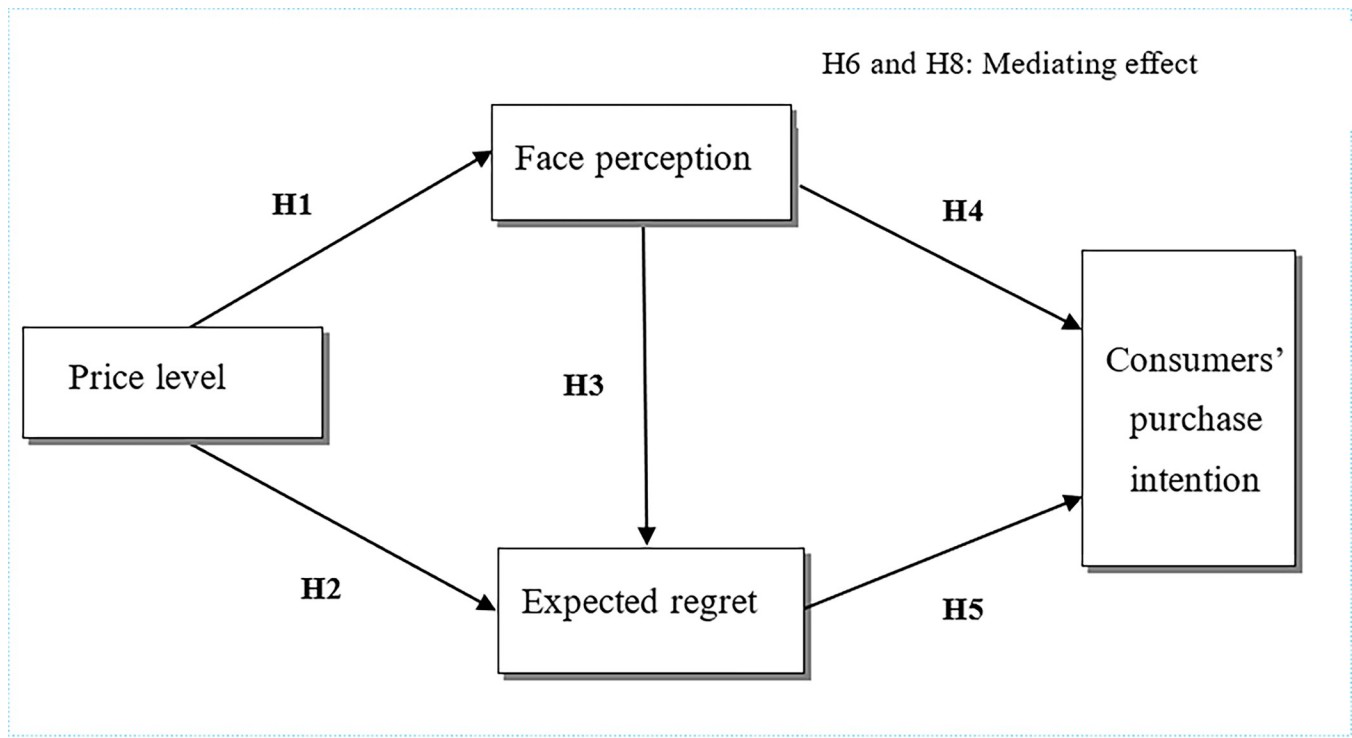

**Figure 1. Research model figure**

**Fig 1. Research model figure.**

questionnaire in a free space to minimize the interference of others on the respondents to fill in the questionnaire and improve the quality of filling in the questionnaire. A total of 357 questionnaires were issued in the formal survey, and 342 questionnaires were recovered. The number of valid questionnaires was 329, excluding incomplete questionnaires and inconsistent answers. The sample composition of the questionnaire is shown in Table 1.

**Table 1. Sample structure.**

| One class indicators | Two class indicators | sample size | Percentage | Oneclass indicatos | Two class indicators | Sample size | Percentage |
|---|---|---|---|---|---|---|---|
| Gender | Male | 161 | 48.93% | Education | Secondary school and below | 41 | 12.46% |
| | Female | 168 | 51.07% | | College degree | 98 | 29.79% |
| Monthly income | Below RMB10000 | 47 | 14.29% | | Bachelor | 129 | 39.21% |
| | RMB10001-20000 | 45 | 13.68% | | Master and doctor | 61 | 18.54% |
| | RMB20001-35000 | 69 | 20.97% | Occupation | Various staff | 91 | 27.66% |
| | RMB35001-50000 | 96 | 29.18% | | Various management personnel | 117 | 35.56% |
| | Above RMB50000 | 72 | 21.88% | | Entrepreneur | 65 | 17.76% |
| Age | 18–25 year | 83 | 25.23% | | Civil service staff | 21 | 6.38% |
| | 26–39 years | 129 | 39.21% | | Student | 18 | 5.47% |
| | 40–59 years | 96 | 29.17% | | Farmer | 9 | 2.73% |
| | Over 60 years | 21 | 6.38% | | Other | 8 | 2.43% |

It should be noted that the supporting information (Minimum data) of this study is the original data from our questionnaire survey (uploaded to the author submission system). This study conducted descriptive analysis of sample structure, reliability and validity test, hypothesis test, and mediation effect test based on supporting information(Minimum data).

### 3.3 Ethics statement

In the questionnaire of this study, the respondents come from the adult citizens of Guilin, China. The researcher obtained verbal consent from the respondents. Researchers allow respondents to fill out questionnaires in a free space to minimize interference from others when filling out questionnaires. According to the regulations of Guilin University of Technology, ethical approval is not required for the questionnaire survey of this study.

## 4. Test and analysis of research data

### 4.1 Reliability and validity

The reliability and validity of this study were tested by SPSS 25.0 and AMOS 25.0. Table 1 shows that Cronbach's α for each variable is 0.744–0.819, reaching Cronbach's α is higher than the standard of 0.7 [27,28], so the questionnaire data of this study passed the reliability test.

The evaluation criteria of convergence validity are that the standardized load coefficient of each test item is greater than 0.50 and reaches statistical significance, the combined reliability (CR) value is greater than 0.70, and the average extraction variance (AVE) is greater than 0.50; The evaluation criterion of discriminant validity is that the square root value of AVE of each variable is not lower than the correlation coefficient between this variable and other variables [29,30]. As shown in Table 1, the standardized load coefficients of the 14 items are above 0.50, the T values are greater than 1.96, the combined reliability (CR) values of the 4 variables are greater than 0.70, and the AVE is greater than 0.50, indicating that the convergence validity has passed the test. As shown in Table 2, the square root value of the AVE of the four variables is greater than the correlation coefficient value between the variable and other variables, so the discriminant validity test. The results of the construction validity test of this study are shown in Table 3. All indicators meet the standard of a good model [29,31] indicating that the model and data are well fitted, so the construction validity passed the test.

**Table 2. Test of reliability and validity.**

| Variables | Measurement items | Load factor | T | Cronbach's α | CR | AVE |
|---|---|---|---|---|---|---|
| Price level | 1.I think this product is expensive | 0.762 | 6.512 | | | |
| | 2.I think the price of this product is high | 0.827 | 4.207 | | | |
| | 3. I think the price of this product is appropriate | 0.755 | 4.475 | 0.761 | 0.844 | 0.575 |
| | 4. I think the price of this product is low | 0.683 | 5.942 | | | |
| Face perception | 5.I hope everyone thinks I can do what ordinary people can't | 0.803 | 2.718 | 0.844 | 0.879 | 0.644 |
| | 6.I care about the admiration and praise of others | 0.805 | 5.465 | | | |
| | 7.I hope to be considered generous, fashionable and avant-garde | 0.766 | 4.683 | | | |
| | 8.I am reluctant to be considered unable to afford this product | 0.835 | 3.287 | | | |
| Expected regret | 9.If you don't buy it, it will be more expensive later | 0.885 | 4.909 | 0.923 | 0.924 | 0.802 |
| | 10.If not buy it, other products are expensive | 0.934 | 3.652 | | | |
| | 11.If you buy it, the price reduction will cause money loss in the future | 0.896 | 3.857 | | | |
| Consumers' purchase intention | 12.I will buy this product | 0.891 | 7.628 | 0.810 | 0.885 | 0.721 |
| | 13.The price will not affect my purchase of the product | 0.860 | 4.734 | | | |
| | 14.I will recommend others to buy this product | 0.793 | 5.268 | | | |

**Table 3. Research model fit.**

| Fit index | χ2 | df | χ2 /df | CFI | TLI | SRMR | RMSEA |
|---|---|---|---|---|---|---|---|
| Index value | 221.794 | 56 | 3.961 | 0.942 | 0.939 | 0.037 | 0.058 |

## 4.2 Research hypothesis test

In order to avoid multicollinearity problems, researchers conducted a centralized analysis of independent variables before the hypothesis test. The data analysis results show that the variance inflation factor (VIF) is between 3.726 and 6.103, which is significantly lower than the threshold value of 10. Therefore, there is no multicollinearity problem in the research model.

In this study, structural equation modeling is used to analyze the internal influence relationship among the measurement variables. Because structural equation model can include multiple dependent variables at the same time, regression analysis will ignore the existence and influence of other dependent variables when analyzing the influence relationship on a certain dependent variable [28]. The structural equation model deals with the comparison of the overall model (nested model), so the reference index is not a single parameter as the main consideration, but an integrated coefficient. Structural equation model analysis can provide different evaluation indicators, which is conducive to analysis from different perspectives and avoid excessive reliance on a single indicator [32].

This study uses AMOS25.0 software and structural equation to verify the research hypothesis. The research results are shown in Table 4. Table 4 shows that except for research hypothesis H2, the test results of all research hypotheses support the original hypothesis. The research results show that the standardized path coefficient of luxury price level to face perception is 0.818, T = 5.457, and the research hypothesis H1 passes the test. In the H2 test, the standardized path coefficient of luxury price level to expected regret is 0.394, T = 1.785, and the research hypothesis does not pass the test, indicating that the increase in price may not bring obvious expected regret behavior to customers. In H3 test, the standardized path coefficient of face perception to expected regret is -0.582, T = 2.518, and its research hypothesis passes the test, indicating that the stronger the consumer's face perception is, the lower the expected regret will be. The H4 test results show that the standardized path coefficient of face perception on purchase intention is 0.761, T = 5.121, and H4 passed the test, indicating that face perception has a significant positive impact on consumer's purchase intention, that is, the stronger the consumer's face perception, the stronger their purchase intention. There are two cases in the test of hypothesis H5. As shown in Table 5, in the H5a test, the standardized path coefficient of consumers' downward expected regret to purchase intention is 0.764, T = 2.278, so

**Table 4. Research hypothesis test.**

| Research hypothesis | Relationship path | Normalized path coefficient | T | Test result |
|---|---|---|---|---|
| H1 | Price level → Face perception | 0.818 | 5.457*** | Support |
| H2 | Price level → Expected regret | 0.394 | 1.785 | Not support |
| H3 | Face perception → Expected regret | -0.582 | 2.518* | Support |
| H4 | Face perception →Consumers' purchase intention | 0.761 | 5.121** | Support |
| H5a | Consumers' downward expected regret →Consumers' purchase intention | 0.764 | 4.278** | Support |
| H5b | Consumer's upward expected regret →Consumers' purchase intention | -0.577 | 3.872** | Support |

Note

* represents p < 0.05

* * represents p < 0.01, and * * represents<0.001.

**Table 5. Correlation coefficient and square root of AVE.**

| Variables | 1 | 2 | 3 | 4 |
|---|---|---|---|---|
| 1. Price level | 0.758 | | | |
| 2. Face perception | 0.663 | 0.802 | | |
| 3. Expected regret | 0.615 | -0.671 | 0.896 | |
| 4. Consumers' purchase intention | -0.391 | 0.716 | 0.675 | 0.849 |

Note: The value on the diagonal is the square root of AVE, and other data are the correlation coefficient values between corresponding variables.

H5a passes the test. This shows that consumers' downward expected regret (if they don't buy, they will regret) will increase their purchase intention, which is due to consumers' expectations and worries about the continued price rise of luxury goods. In the H5b test, the standardized path coefficient of consumers' upward expected regret to purchase intention is -0.577, T = 3.872, and H5b also passes the test, which shows that if consumers think "they will regret if they buy", they will reduce their purchase intention.

### 4.3 Mediating effect test

This study involves mediating effects hypothesis H6, H7 and H8, which are simple mediating effects. Preacher et al. believed that the Bootstrap method should be used for the simple mediating effect test [32]. In terms of the operation method of Bootstrap method, this study draws on the practice of Chen et al. [33], Ma & Zhou [34], use Model 6 in the PROCESS macro prepared to test the mediating effect of H6, H7 and H8 [35]. The operation process is as follows: Install the PROCESS macro into SPSS 25.0, then run SPSS25.0, select the menu "Analyze Expression Process", select the operation variables (independent variables, mediation variables and dependent variables) into the option box, select 4 in the menu of "Model Number", set "Bootstrap Samples" to 5000 times, select "Bia Corrected" for the sampling method, set the confidence interval to 95%, and finally click "OK", and complete the operation of using SPSS 25.0 to test the mediating effect.

It can be seen from Table 6 that the model fitting indicators of the competition model confirm that it is not a good model, while the indicators of the chain mediating effect model meet the criteria of a good model, so the chain mediating effect does exist. In Table 7, the indirect effect value of "price level → face perception → consumers' purchase intention" is equal to 0.151, the indirect effect value of "price level → expected regret → consumers' purchase intention" is equal to 0.147, and the indirect effect value of "price level → face perception → expected regret → consumers' purchase intention" is equal to 0.174. Therefore, the mediating effect research hypothesis H6, H7 and H8 passed the test.

## 5.Conclusions and discussion

### 5.1 Conclusion and theoretical significance

First, this study shows that the price level of luxury goods has a positive impact on consumers' perception of face, while the positive impact of luxury goods price level on consumers'

**Table 6. Comparison of two models.**

| Model | $\chi^2$ | df | $\chi^2$ /df | CFI | TLI | SRMR | RMSEA |
|---|---|---|---|---|---|---|---|
| Chain mediating model | 43.146 | 14 | 3.081 | 0.928 | 0.930 | 0.045 | 0.072 |
| Competition model | 107.672 | 16 | 6.729 | 0.738 | 0.779 | 0.077 | 0.119 |

**Table 7. Test results of mediating effect.**

| Mediating effect path | Indirect effect value | Standard error | Upper limit | Lower limit | Effect proportion |
|---|---|---|---|---|---|
| 1. Price level → face perception → consumers' purchase intention | 0.151 | 0.002 | 0.311 | 0.102 | 21.45% |
| 2. Price level → expected regret → consumers' purchase intention | 0.147 | 0.013 | 0.305 | 0.099 | 20.88% |
| 3. Price level → face perception → expected regret → consumers' purchase intention | 0.174 | 0.007 | 0.336 | 0.108 | 24.72% |
| 4. Total mediating effect | 0.472 | 0.015 | 0.639 | 0.299 | 67.05% |
| 5. Total effect | 0.704 | 0.018 | 0.892 | 0.398 | 100% |

expected regret has not been confirmed, indicating that price increases may not necessarily lead to significant expected regret behavior among consumers. In addition, this study also confirmed that the stronger the consumers' face perception, the lower the consumers' expected regret. The above conclusions of this study are the inheritance and innovation of Ji et al [8] Yang et al [14]., and Ye et al [19]. Therefore, this study explores the study of luxury purchase intention from the perspectives of face saving and expected regret. There is still little research on this topic, which will help enrich the content of luxury purchase intention research and provide reference and reference for future related research.

Second, this study shows that price level of luxury goods has a positive impact on consumers' face perception, while the positive impact of the price level of luxury goods on consumers' expected regret has not been confirmed, indicating that rising prices may not necessarily lead to significant expected regret behavior. In previous studies, Wang et al. confirmed that consumers with low self-esteem will have more obvious impulse buying behavior after gaining face [24]. Consumer face awareness will have a little impact on purchase intention, but previous studies have little research on the impact of expected regret in purchase intention [1]. Therefore, this study explores the impact of consumers' expected regret on their willingness to purchase luxury goods, which will expand the theoretical application of expected regret and willingness to purchase, and enrich the connotation of luxury goods consumption research.

Third, in terms of the mediating effect test, this study confirmed that face perception played a mediating effect between price level and purchase intention, and expected regret played a mediating effect between price level and purchase intention. In addition, in the relationship between price level and consumers' purchase intention, face perception and expected regret play a chain mediating effect. Compared with previous studies, this study is the development of research conclusions of Yin & Yu [5], Zhang & Zhuang [25]. Because in previous studies, although the impact of relevant variables on consumers' purchase intention has been discussed, there is still a lack of attention to the mediating effect in the impact relationship. This study explores the mediating effect of face perception and expected regret, which is conducive to better analyzing the impact mechanism of luxury goods purchase intention, integrating psychological theory and marketing theory, and can better explain consumer psychology and behavior problems in luxury goods consumption.

## 5.2 Practical significance

First, businesses should implement scientific and effective pricing strategies tailored to the high price elasticity of luxury goods. Since 2023, the mobility of people across China has increased, and the Chinese market has become active, with many consumers willing to purchase luxury goods. Therefore, businesses need to seize this opportunity and expand sales of luxury goods. According to modern economic theory, price elasticity reflects the ratio of supply and demand changes caused by price changes, which is the sensitivity level of supply and demand to price information. Luxury goods are commodities with price elasticity, and small

changes in their prices will cause large changes in demand. Based on the price elasticity of luxury goods, luxury goods merchants should make a small reduction in the relatively unsalable luxury goods during the holidays and implement the price discount strategy, which will improve the sales volume and thus enhance the profitability of enterprises as a whole.

Second, businesses should respect consumers' face perception, meet their face needs, and stimulate their consumption potential. For a long time, saving face has been an important feature of Chinese culture and a key to gaining a deeper understanding of Chinese society. In China, many consumers who purchase luxury goods are motivated by the need for face and are also symbols of status and status. The uses of luxury goods include gifting, self-use, or collection, all of which involve consumer face. Therefore, when merchants sell luxury goods, they should highlight the nobility and rarity of luxury goods in promotion, which is a symbol of consumer identity and status. Consuming these luxury goods is very prestigious, enhancing consumers' recognition of the noble characteristics of luxury goods, meeting consumers' face needs, and stimulating their purchasing desire.

Third, government departments and businesses should reasonably guide consumers' psychological expectations, conduct scientific consumption, and promote social harmony. Expected regret is a psychological problem that consumers often encounter during the consumption process. Consumers often regret their purchases, which can affect their future purchasing intentions and word-of-mouth. The government should provide scientific and reasonable guidance on luxury consumption through public media (such as television, Weibo, newspapers, etc.), promote "acting within one's capabilities and moderately leading", oppose excessive leading consumption, especially through loans for luxury consumption, and scientifically manage one's own consumption expectations. Governments and businesses can invite economic, marketing and psychological experts to analyze, answer and guide consumers' tangled "regret when buying" and "regret when not buying" through TV, Tiktok, newspapers, microblog, etc., and scientifically guide consumers' psychological expectations.

## 5.3 Limitations and prospects

First, in terms of research content. This study did not include cognitive dissonance, perceived risk, and emotional contagion in the research content. These variables may have a certain impact on face perception and purchase intention, or cognitive dissonance, perceived risk, and other variables may play a moderating role between price level and purchase intention. In order to improve the scientific nature of the research, I will conduct research hypotheses and empirical studies on the relationship between the above variables, extending the current research.

Second, in terms of investigating sample sources. Our survey sample comes from Guilin City, China, which is a tourist city in Guangxi, China. The respondents include people from various professions. However, China has a vast territory and a large population. In future research, we will strive to conduct questionnaire surveys or scenario experiments in multiple cities in China to obtain research data. In addition, we also plan to conduct relevant sample surveys in countries and regions such as East Asia, Southeast Asia, and Europe, to broaden our research horizons.

Third, in terms of research methods. The questionnaire survey method used in this study collected research data, but the questionnaire survey data were all derived from the subjective evaluation of the respondents, who often had certain cognitive biases and were influenced by the emotions and situations at the time. Therefore, in future related research, we will also obtain the objective data of consumers on online shopping platforms (such as Taobao, JD.com, Pinduoduo, etc.) through the web crawler method, including the time and times of

consumers logging into luxury online stores, as well as the number of purchases. In addition, we can also obtain research data through scenario experiments, innovate research methods, and make research conclusions more reliable and have promotional value.

## Supporting information

**S1 Dataset.**
(SAV)

## Author Contributions

**Conceptualization:** Jiahua Wei.

**Data curation:** Jiahua Wei.

**Formal analysis:** Jiahua Wei.

**Funding acquisition:** Jiahua Wei.

**Investigation:** Jiahua Wei, Hong Shen, Zhizeng Lu.

**Methodology:** Jiahua Wei, Hong Shen, Zhizeng Lu.

**Project administration:** Jiahua Wei.

**Software:** Jiahua Wei.

**Supervision:** Hong Shen.

**Visualization:** Jiahua Wei.

**Writing – original draft:** Jiahua Wei.

**Writing – review & editing:** Jiahua Wei, Hong Shen, Zhizeng Lu.

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
