## [Decision Letter · Decision Letter 0]

3 Nov 2023

PONE-D-23-23447A Study on the Purchase Intention of Luxury Goods from the Perspective of Face Perception and Expected Regret in the Chinese ContextPLOS ONE

Dear Dr. Wei,

Thank you for submitting your manuscript to PLOS ONE. After careful consideration, we feel that it has merit but does not fully meet PLOS ONE’s publication criteria as it currently stands. Therefore, we invite you to submit a revised version of the manuscript that addresses the points raised during the review process.

We look forward to receiving your revised manuscript.

Kind regards,

Alhamzah F. Abbas, PhD

Academic Editor

PLOS ONE

Journal Requirements:

2. You indicated that ethical approval was not necessary for your study. We understand that the framework for ethical oversight requirements for studies of this type may differ depending on the setting and we would appreciate some further clarification regarding your research. Could you please provide further details on why your study is exempt from the need for approval and confirmation from your institutional review board or research ethics committee (e.g., in the form of a letter or email correspondence) that ethics review was not necessary for this study? Please include a copy of the correspondence as an "Other file".

Also please provide additional details regarding participant consent. In the ethics statement in the Methods and online submission information, please ensure that you have specified (1) whether consent was informed and (2) what type you obtained (for instance, written or verbal, and if verbal, how it was documented and witnessed). If your study included minors, state whether you obtained consent from parents or guardians. If the need for consent was waived by the ethics committee, please include this information.

5. We note you have included a table to which you do not refer in the text of your manuscript. Please ensure that you refer to Table 5 in your text; if accepted, production will need this reference to link the reader to the Table.

Reviewers' comments:

Reviewer's Responses to Questions

**Comments to the Author**

1. Is the manuscript technically sound, and do the data support the conclusions?

Reviewer #1: No

Reviewer #2: Yes

2. Has the statistical analysis been performed appropriately and rigorously? 

Reviewer #1: I Don't Know

Reviewer #2: Yes

3. Have the authors made all data underlying the findings in their manuscript fully available?

Reviewer #1: No

Reviewer #2: No

4. Is the manuscript presented in an intelligible fashion and written in standard English?

Reviewer #1: No

Reviewer #2: Yes

5. Review Comments to the Author

Reviewer #1: 1. The study collected data through a questionnaire survey in Guilin, China. However, the size of the sample and its representativeness in reflecting the broader population of luxury goods consumers in China could be a concern. A small or non-diverse sample might limit the generalizability of the findings.

2. The study identifies relationships between variables like price level, face perception, expected regret, and purchase intention. However, it's important to establish whether these relationships are causal or merely correlated. The study should provide a strong rationale for the proposed causal relationships and consider alternative explanations.

3. The results of the study might be specific to the local context of Guilin, China, and may not be applicable to luxury consumers in other regions with different socioeconomic, cultural, and economic conditions. The study should discuss the external validity and potential limitations in applying the findings more broadly.

4. The study relies on self-reported data collected through questionnaires. The accuracy and reliability of respondents' answers might be influenced by factors like social desirability bias, memory recall issues, and interpretation of questions.

5. The study suggests that consumer face perception and expected regret mediate the relationship between price level and purchase intention. However, the actual mechanisms behind these mediations should be thoroughly examined, and potential third variables (moderators) that might influence these relationships should be considered.

6. The study focuses on the post-liberalization period after COVID-19 control measures. However, luxury goods consumption behaviors might change over time due to evolving economic, social, and cultural factors.

7. While the study aims to contribute to the understanding of consumer psychology and behavior, it's important to assess the strength of the theoretical framework used. Are there alternative theories or models that could explain the observed relationships?

8. The study claims to promote the healthy development of the luxury goods industry. However, the specific practical implications for industry stakeholders (e.g., marketers, policymakers) should be outlined more explicitly based on the study's findings.

Reviewer #2: The manuscript presents an intriguing approach to understanding the factors influencing the purchase intentions of luxury goods consumers, focusing on face perception and expected regret. Methodologically, the paper appears sound, with a clear empirical strategy and analysis. However, there are several concerns regarding the breadth and depth of the literature review, as well as the current relevance of COVID-19 to the model.

1. The literature review predominantly cites studies from Chinese authors, which may inadvertently introduce a geographic bias. It is recommended that the author(s) incorporate a more diverse range of literature, including seminal and contemporary works from various regions that have explored the determinants of luxury goods consumption.

2.

The title currently restricts the study to the Chinese context, which could limit the generalizability of the findings.

3. The paper would benefit from an expanded review that includes studies addressing factors affecting consumer decisions beyond the current variables of price level, face perception, and expected regret. Such factors might include cultural influences, marketing communication, brand reputation, and consumer values, which are known to impact luxury purchase intentions. Certainly, here are the references with author names and DOI information removed:. "The influence of green finance availability to retailers on purchase intention: a consumer perspective with the moderating role of consciousness." Environmental Science and Pollution Research, 1-17."Influence of virtual CSR co-creation on the purchase intention of green products under the heterogeneity of experience value." Sustainability, 14(20), 13617."Managing skill certification in online outsourcing platforms: A perspective of buyer-determined reverse auctions." International Journal of Production Economics, 238."Exploring changes in guest preferences for Airbnb accommodation with different levels of sharing and prices: Using structural topic model." Frontiers in psychology, "The effect of image enhancement on influencer's product recommendation effectiveness: the roles of perceived influencer authenticity and post type." Journal of Research in Interactive Marketing.

4. Given the submission date and the decline in the immediate impact of the COVID-19 pandemic, the references to it might be outdated. Unless the pandemic's influence is directly examined as part of the model, it would be advisable to omit these references. The study should focus on the enduring aspects of consumer behavior in luxury goods markets.

5. Where is structure model figure?

6. English needs improvement , proofreading.

6. PLOS authors have the option to publish the peer review history of their article (what does this mean?). If published, this will include your full peer review and any attached files.

Reviewer #1: No

Reviewer #2: No

---

## [Author Response · Author response to Decision Letter 0]

20 Nov 2023

Response to Reviewers

Dear reviewers,

We write in response to the modifications made to the manuscript “A Study on the Purchase Intention of Luxury Goods from the Perspective of Face Perception and Expected Regret” submitted for publication. Based on the reviewer’ recommendations made by the reviewers, the manuscript has been revised taking cognizance of the insightful comments. All revisions (include additional references) in the manuscript are marked with red color. The comments (in red color) by the reviewers and the response (in blue) by authors are found below.

 We would like also to thank you for allowing us to resubmit a revised copy of the manuscript. We hope that the revised manuscript is accepted for publication in the PLOS ONE.

Response to reviewers’ comments

Reviewer: #1: 

1.The study collected data through a questionnaire survey in Guilin, China. However, the size of the sample and its representativeness in reflecting the broader population of luxury goods consumers in China could be a concern. A small or non-diverse sample might limit the generalizability of the findings.

Reply:

Thank you very much for the review expert's question. I am happy to explain your question. Firstly, according to the sample size requirements of the questionnaire survey, Wu (2010) believes that when the sample size is more than 10 times the number of questionnaire items, the sample size is considered sufficient. The effective sample size for the questionnaire survey in this study is 329, and the questionnaire items are 14. The sample size is 23.5 times that of the items, so our sample size is sufficient. Secondly, I chose Guilin in China for sample collection, mainly considering that Guilin is an ordinary city in China, with a certain representativeness in terms of population size and economic development level. Thank you again for the review expert's question.

2.The study identifies relationships between variables like price level, face perception, expected regret, and purchase intention. However, it's important to establish whether these relationships are causal or merely correlated. The study should provide a strong rationale for the proposed causal relationships and consider alternative explanations.

Reply: 

Thank you very much for the review expert’s question. We explored the relationship between variables such as price level, face perception, expected regret, and purchase intention in our study, which is a causal relationship. We followed the general writing method of empirical research papers and first proposed research hypotheses between several variables. Then, we conducted a questionnaire survey and analyzed the validity and reliability of the questionnaire data. On this basis, this study used AMOS25.0 software to validate the research hypothesis through structural equations. Table 5 shows the causal relationship between each variable (P12). In addition, this study provides an explanation of the research results(P12), and also discusses the research conclusions (P13-P15).

3. The results of the study might be specific to the local context of Guilin, China, and may not be applicable to luxury consumers in other regions with different socioeconomic, cultural, and economic conditions. The study should discuss the external validity and potential limitations in applying the findings more broadly.

Reply:

Thank you very much for the review expert’s question. We also discussed this issue in “5.3. Research Limitations and Prospects” of the revised manuscript. We believe that Guilin is a representative city in China, and many studies have also selected a representative city and a representative group for research. There are actually certain similarities in consumer behavior among different regions. Especially in the same Chinese cultural background, consumers from different cities have familiar cultural cognition. Of course, in future research, we will also strive to conduct questionnaire surveys or scenario experiments in multiple cities in China to obtain research data. In addition, we also plan to conduct relevant surveys and research in countries and regions such as East Asia, Southeast Asia, and Europe, to broaden our research horizons.

4. The study relies on self-reported data collected through questionnaires. The accuracy and reliability of respondents' answers might be influenced by factors like social desirability bias, memory recall issues, and interpretation of questions.

Reply:

Thank you for the review expert’s question. Questionnaire surveys do have some shortcomings, but they are also a commonly used and effective method for studying consumer behavior. In “5.3. Research Limitations and Prospects”, we discussed the limitations of the questionnaire survey and our future research plans as follows:

“Third, in terms of research methods. The questionnaire survey method used in this study collected research data, but the questionnaire survey data were all derived from the subjective evaluation of the respondents, who often had certain cognitive biases and were influenced by the emotions and situations at the time. Therefore, in future related research, we will also obtain the objective data of consumers on online shopping platforms (such as Taobao, JD.com, Pinduoduo, etc.) through the web crawler method, including the time and times of consumers logging into luxury online stores, as well as the number of purchases. In addition, we can also obtain research data through scenario experiments, innovate research methods, and make research conclusions more reliable and have promotional value.”

5. The study suggests that consumer face perception and expected regret mediate the relationship between price level and purchase intention. However, the actual mechanisms behind these mediations should be thoroughly examined, and potential third variables (moderators) that might influence these relationships should be considered.

Reply:

Thank you for the review expert’s suggestions. In our study, we empirically confirmed that consumers' face perception and expected regret play a mediating role in the relationship between price level and purchase intention. There may also be a third variable that plays a moderating role between them, as we found in “5.3 The research limitations and prospects” are discussed as follows:

“First, in terms of research content. This study did not include cognitive dissonance, perceived risk, and emotional contagion in the research content. These variables may have a certain impact on face perception and purchase intention, or cognitive dissonance, perceived risk, and other variables may play a moderating role between price level and purchase intention. In order to improve the scientific nature of the research, I will conduct research hypotheses and empirical studies on the relationship between the above variables, extending the current research.”

6. The study focuses on the post-liberalization period after COVID-19 control measures. However, luxury goods consumption behaviors might change over time due to evolving economic, social, and cultural factors.

Reply: 

Thank you for the suggestions from the evaluation experts. This is a practical issue. At present, China has resumed normal economic activity and people's lives are also normal. Therefore, in the revised version, we no longer emphasize the special background of “China has liberalized the prevention and control of the COVID-19 epidemic”. Correspondingly, we have made corresponding modifications to the abstract, introduction, literature review and research hypotheses, conclusions and discussions of the paper. Thank you again for the suggestions from the reviewers.

7. While the study aims to contribute to the understanding of consumer psychology and behavior, it's important to assess the strength of the theoretical framework used. Are there alternative theories or models that could explain the observed relationships?

Reply: 

Thank you for the review expert’s question. The theoretical framework of this study is proposed based on research hypotheses, and our research hypotheses refer to a large number of domestic and foreign literature, including expected regret theory, consumer purchase intention theory, etc. Therefore, our research hypotheses and theoretical framework are not based on imagination, but have theoretical basis. Similar studies have been conducted in some literature on the relationship between variables in our theoretical framework. For example, Yin and Yu (2008) argue that the direction of consumer expected regret has a significant impact on impulsive purchasing behavior. Zhang and Zhuang (2008) argue that informational social influence tendencies indirectly affect impulsive purchases through the mediating variable of face preservation tendencies. Ji et al. (2022) conducted an empirical study on the Jiaodong region of China, which showed that face perception plays a mediating role in the process of self-concept influencing wine purchase intention. Although these studies are not focused on the field of luxury goods, they can also provide important theoretical references for this study. On the other hand, previous research findings can further explain our hypothesis relationships. Thank you again for the suggestions from the reviewers.

8. The study claims to promote the healthy development of the luxury goods industry. However, the specific practical implications for industry stakeholders (e.g., marketers, policymakers) should be outlined more explicitly based on the study's findings.

Reply:

Thank you very much for the suggestions from the reviewers. In the revised manuscript of the paper, in order to discuss the development of the luxury goods industry and its impact on stakeholders, we specifically discussed the following in “5.2 Practical significance” (P15-P16)

“First, businesses should implement scientific and effective pricing strategies tailored to the high price elasticity of luxury goods. Since 2023, the mobility of people across China has increased, and the Chinese market has become active, with many consumers willing to purchase luxury goods. Therefore, businesses need to seize this opportunity and expand sales of luxury goods. According to modern economic theory, price elasticity reflects the ratio of supply and demand changes caused by price changes, which is the sensitivity level of supply and demand to price information. Luxury goods are commodities with price elasticity, and small changes in their prices will cause large changes in demand. Based on the price elasticity of luxury goods, luxury goods merchants should make a small reduction in the relatively unsalable luxury goods during the holidays and implement the price discount strategy, which will improve the sales volume and thus enhance the profitability of enterprises as a whole.

Second, businesses should respect consumers’ face perception, meet their face needs, and stimulate their consumption potential. For a long time, saving face has been an important feature of Chinese culture and a key to gaining a deeper understanding of Chinese society. In China, many consumers who purchase luxury goods are motivated by the need for face and are also symbols of status and status. The uses of luxury goods include gifting, self-use, or collection, all of which involve consumer face. Therefore, when merchants sell luxury goods, they should highlight the nobility and rarity of luxury goods in promotion, which is a symbol of consumer identity and status. Consuming these luxury goods is very prestigious, enhancing consumers’ recognition of the noble characteristics of luxury goods, meeting consumers' face needs, and stimulating their purchasing desire.

Third, government departments and businesses should reasonably guide consumers’ psychological expectations, conduct scientific consumption, and promote social harmony. Expected regret is a psychological problem that consumers often encounter during the consumption process. Consumers often regret their purchases, which can affect their future purchasing intentions and word-of-mouth. The government should provide scientific and reasonable guidance on luxury consumption through public media (such as television, Weibo, newspapers, etc.), promote “acting within one’s capabilities and moderately leading”, oppose excessive leading consumption, especially through loans for luxury consumption, and scientifically manage one’s own consumption expectations. Governments and businesses can invite economic, marketing and psychological experts to analyze, answer and guide consumers' tangled “regret when buying” and “regret when not buying” through TV, Tiktok, newspapers, microblog, etc., and scientifically guide consumers’ psychological expectations.”

Reviewer #2: 

The manuscript presents an intriguing approach to understanding the factors influencing the purchase intentions of luxury goods consumers, focusing on face perception and expected regret. Methodologically, the paper appears sound, with a clear empirical strategy and analysis. However, there are several concerns regarding the breadth and depth of the literature review, as well as the current relevance of COVID-19 to the model.

1. The literature review predominantly cites studies from Chinese authors, which may inadvertently introduce a geographic bias. It is recommended that the author(s) incorporate a more diverse range of literature, including seminal and contemporary works from various regions that have explored the determinants of luxury goods consumption.

Reply:

Thank you very much for the reviewer's suggestion. Based on your suggestion, in the paper revision, we have cited some research from scholars outside of China in the literature review, such as: “Crawford, M. T., Mcconnell, A. R., Lewis, A. C., & Sherman, S. J. (2002). Reactance, compliance, and anticipated regret. Journal of Experimental Social Psychology, 38(1), 56-63”，“Jeon, Hoseong.(2014). The effect of anticipated regret upon consumers’ unplanned purchases. Journal of Consumption Culture.17(2),1-23” .

We also cited 5 references recommended by the reviewers to diversify the literature. Please refer to the revised references (P17-P20).

2.The title currently restricts the study to the Chinese context, which could limit the generalizability of the findings.

Reply:

Thank you very much for the suggestions from the reviewers. In the paper revision, we revised the title to “A Study on the Purchase Intention of Luxury Goods from the Perspective of Face Perception and Expected Regret”. Although we conducted empirical research based on the Chinese sample, it also has theoretical and practical guidance for other countries and regions. Thank you again for the expert’s advice.

3.The paper would benefit from an expanded review that includes studies addressing factors affecting consumer decisions beyond the current variables of price level, face perception, and expected regret. Such factors might include cultural influences, marketing communication, brand reputation, and consumer values, which are known to impact luxury purchase intentions. Certainly, here are the references with author names and DOI information removed:. "The influence of green finance availability to retailers on purchase intention: a consumer perspective with the moderating role of consciousness." Environmental Science and Pollution Research, 1-17."Influence of virtual CSR co-creation on the purchase intention of green products under the heterogeneity of experience value." Sustainability, 14(20), 13617."Managing skill certification in online outsourcing platforms: A perspective of buyer-determined reverse auctions." International Journal of Production Economics, 238."Exploring changes in guest preferences for Airbnb accommodation with different levels of sharing and prices: Using structural topic model." Frontiers in psychology, "The effect of image enhancement on influencer's product recommendation effectiveness: the roles of perceived influencer authenticity and post type." Journal of Research in Interactive Marketing.

Reply: Thank you very much for the suggestions and recommendations of the experts. We downloaded these papers, carefully read them, and cited them in the revised manuscript. By referring to these papers, we have expanded our research horizons, especially in the study of consumer purchase intention. Please refer to P17-P20 for the citation information of the above papers. Thank you again for your professional suggestions.

4. Given the submission date and the decline in the immediate impact of the COVID-19 pandemic, the references to it might be outdated. Unless the pandemic's influence is directly examined as part of the model, it would be advisable to omit these references. The study should focus on the enduring aspects of consumer behavior in luxury goods markets. 

Reply: 

Thank you very much for the suggestions from the reviewers. We believe that the impact of the COVID-19 pandemic has indeed decreased and is no longer a concern for society. The social order in China has also been restored. Therefore, in the revised manuscript, we have removed the content related to the COVID-19 pandemic, such as deleting the first paragraph of the introduction and modifying the abstract, introduction, and conclusion. Thank you again for your suggestion.

5. Where is structure model figure?

Reply:

Dear reviewer, our structural model diagram is shown in P7, as shown in Figure 1. We refer to it as the research model in our paper. Based on the research hypothesis of this study, the research model describes the influence relationship between the four research variables, including the mediating effect relationship.

6. English needs improvement , proofreading.

Reply: 

Thank you very much for the suggestions from the reviewers. Based on your suggestion, we have made modifications and refinements to the English language expression during the paper process, striving to improve the accuracy of language expression. 

Please review our revised paper. Thank you again for your comments and guidance.

---

## [Editor Report · Decision Letter 1]

28 Dec 2023

A Study on the Purchase Intention of Luxury Goods from the Perspective of Face Perception and Expected Regret

PONE-D-23-23447R1

Dear Dr. Jiahua Wei,

We’re pleased to inform you that your manuscript has been judged scientifically suitable for publication and will be formally accepted for publication once it meets all outstanding technical requirements.

Kind regards,

Alhamzah F. Abbas, PhD

Academic Editor

PLOS ONE
---

## [Editor Report · Acceptance letter]

14 Mar 2024

PONE-D-23-23447R1 

PLOS ONE

Dear Dr. Wei, 

I'm pleased to inform you that your manuscript has been deemed suitable for publication in PLOS ONE. Congratulations! Your manuscript is now being handed over to our production team.

Kind regards, 

on behalf of

Dr. Alhamzah F. Abbas 

Academic Editor

PLOS ONE